# Achieving HIV epidemic control through integrated community and facility-based strategies: Lessons learnt from ART-surge implementation in Akwa Ibom, Nigeria

Pius Nwaokoro[1]☉*, Olusola Sanwo[1]☉, Otoyo Toyo[2]☉, Uduak Akpan[2]☉, Esther Nwanja[2]☉, Iheanyichukwu Elechi[2]☉, Kufre-Abasi Ukpong[2], Helen Idiong[2], Bala Gana[2], Titilope Badru[1], Augustine Idemudia[2], Matthew-David Ogbechie[1], Philip Imohi[1], Anthony Achanya[3], Dorothy Oqua[3], Kunle Kakanfo[4], Kolawole Olatunbosun[1], Augustine Umoh[5], Patrick Essiet[5], Ime Usanga[5], Echezona Ezeanolue[6], Chika Obiora-Okafo[4], Ezekiel James[4], Isa Iyortim[4], Robert Chiegil[7], Hadiza Khamofu[1], Satish Raj Pandey[1], Moses Bateganya[7]

**1** Family Health International, Abuja, Nigeria, **2** Achieving Health Nigeria Initiative (AHNi), Abuja, Nigeria, **3** Howard University Global Initiative, Abuja, Nigeria, **4** USAID, Abuja, Nigeria, **5** Ministry of Health, Uyo, Nigeria, **6** Center for Translation & Implementation Research College of Medicine, University of Nigeria, Nsukka, Nigeria, **7** Family Health International, Durham, NC, United States of America

☉ These authors contributed equally to this work.
* dr.pcnwaokoro@gmail.com

**Data Availability Statement:** All relevant data are within the paper and its Supporting Information files.

## Abstract

This study examines the lessons learnt from the implementation of a surge program in Akwa Ibom State, Nigeria as part of the Strengthening Integrated Delivery of HIV/AIDS Services (SIDHAS) Project. In this analysis, we included all clients who received HIV counseling and testing services, tested HIV positive, and initiated ART in SIDHAS-supported local government areas (LGAs) from April 2017 to March 2021. We employed descriptive and inferential statistics to analyze our results. A total of 2,018,082 persons were tested for HIV. Out of those tested, 102,165 (5.1%) tested HIV-positive. Comparing the pre-surge and post-surge periods, we observed an increase in HIV testing from 490,450 to 2,018,082 (p≤0.031) and in HIV-positive individuals identified from 21,234 to 102,165 (p≤0.001) respectively. Of those newly identified positives during the surge, 98.26% (100,393/102,165) were linked to antiretroviral therapy compared to 99.24% (21,073/21,234) pre-surge. Retention improved from 83.3% to 92.3% (p<0.001), and viral suppression improved from 73.5% to 96.2% (p<0.001). A combination of community and facility-based interventions implemented during the surge was associated with the rapid increase in case finding, retention, and viral suppression; propelling the State towards HIV epidemic control. HIV programs should consider a combination of community and facility-based interventions in their programming.

**Funding:** This publication resulted in part from data collected during the implementation of the PEPFAR-funded SIDHAS project in Nigeria (Cooperative Agreement Number: AID-620-A-11-00002). The funders had no role in study design, data collection and analysis, decision to publish, or preparation of the manuscript.

**Competing interests:** The authors have declared that no competing interests exist.

**Abbreviations:** ART, Antiretroviral Therapy; CAM, Community ART management; CARC, Community ART Refill club; CARG, Community ART Refill group; CPARP, Community Pharmacy ART Refill Program; DATIM, Data for accountability, integrity, and management; DEC, Data Entry Clerk; DBS, Dry-blood spot; DHIS, District Health Information System; EMR, Electronic Medical records; EAC, Enhanced adherence counseling; FHI 360, Family Health International; HTS, HIV testing services; ICT, Index case testing; LAMIS, Lafiya Management Information System; MSF, Monthly Summary form; NAIIS, Nigeria HIV/AIDS Indicator and Impact Survey; PEPFAR, President's Emergency Plan for AIDS Relief; PITC, Provider initiated testing and counseling; PMTCT, Prevention of mother-to-child transmission of HIV/AIDS; SIDHAS, Strengthening Integrated Delivery of HIV/AIDS Services; UNAIDS, Joint United Nations Programme on HIV/AIDS; USAID, United States Agency for International Development; U = U, Undetectable equals untransmittable; VL, Viral Load; V-DOT, Virtual–Direct Observed Therapy.

# Introduction

On November 18, 2014, the Joint United Nations Programme on HIV/AIDS (UNAIDS) adopted the fast-track targets to end the AIDS epidemic globally by 2030. Its objective was to ensure that 95% of all people living with HIV (PLHIV) know their HIV status, 95% of those diagnosed with HIV infection were on sustained antiretroviral therapy (ART), and 95% of those on ART are virally suppressed [1]. In Nigeria, the journey towards meeting the UNAIDS targets and ending the AIDS epidemic was initially limited by the availability of reliable and actionable data [2, 3] with HIV prevalence data mostly coming from antenatal care (ANC) sentinel surveys or the National HIV/AIDS and Reproductive Health Survey (NARHS) [4] which often overestimated prevalence (10.8% and 6.5% respectively).

The 2018 Nigeria HIV/AIDS Indicator and Impact Survey (NAIIS), used more robust methods to provide incidence and prevalence data for Nigeria, giving a clearer understanding of Nigeria's HIV epidemic [5, 6]. NAIIS showed a 1.4% prevalence of HIV nationally, with an estimated 1.9 million Nigerians living with HIV (2019 spectrum estimate). Among adults aged 15–69 years, Akwa Ibom State had the highest HIV prevalence at 5.5% [5, 7] representing 178,051 people but only 23% were on ART [8]. The President's Emergency Plan for AIDS Relief (PEPFAR) Nigeria team used these findings to launch the Nigeria treatment surge plan which realigned PEPFAR programming and resources to rapidly increase access to ART to ensure that high HIV burden states quickly attain treatment saturation. Thus, Akwa Ibom and Rivers States, two states with the highest HIV burden, lowest testing coverage, and lowest population viral suppression were designated as "surge" states. Other states were designated as "red" states (low treatment saturation and high unmet need for ART), "yellow" (low treatment saturation and low unmet need), and "green" (high treatment saturation and low unmet need) [8].

The United States Agency for International Development (USAID) launched the ART surge in Akwa Ibom State in April 2019 based on these epidemic dynamics. This paper describes that surge and its impact on HIV case identification and treatment outcomes.

# Materials and methods

## Study setting

Akwa Ibom State is in the southern part of Nigeria and has an estimated population of 5.4 million [9]. There is a mixture of urban and metropolitan communities in the north engaged in institutional learning or commercial activities, riverine communities with inland waterways, creeks, and predominantly fisherfolk in the east, agrarian communities in the west, deep coastal areas and islands in the south, with numerous hard-to-reach areas [10].

The surge was initially implemented by the SIDHAS Project in all 31 local government areas (LGAs) in the state for the first six months (April 2019 –Sept 2019), and then in 21 LGAs from October 2019 to March 2021 due to reduction in the geographical scope of the project. One hundred and two health facilities (1 tertiary, 20 secondary, 15 private-for-profit, and 66 primary health facilities), 45 community pharmacies, and 73 community ART management (CAM) teams were involved all through the implementation.

## Study population

All clients who received HIV counseling and testing services and those receiving care and treatment services in the SIDHAS supported LGAs from April 2017 to March 2021.

## The surge intervention

**Surge framework.** Although the surge response was not designed as research, our surge programming was guided by the Exploration, Preparation, Implementation, and Sustainment (EPIS) framework [11]. The "ART surge" intervention involved the implementation of innovative program approaches to overcome structural and institutional barriers and rapidly scale up access to HIV testing, ART, and viral load (VL) testing. To increase HIV case-finding rates, community mobilization, deployment of multidisciplinary teams, and use of HIV risk stratification tools were employed [12].

Individuals identified as HIV-positive were promptly linked to ART and offered client-centric care (such as convenient appointment schedules, decentralized drug pick-up, virtual adherence support, etc) with a robust client support system. VL samples were collected in the community and at health facilities and transported to testing hubs for rapid processing. Viral Load results were promptly returned to providers through the online Laboratory Information Management System (LIMS). This platform allows service providers to remotely log-in VL samples to the reference laboratory, receive, and print results once analysis is complete. Data on key indicators such as the number of newly diagnosed PLHIV were reported daily (high-frequency reporting), aggregated per site/LGA, and discussed in daily situation room meetings and technical assistance was provided based on need.

Fig 1 shows how we adapted the EPIS Framework utilizing key informant interviews with community leaders to capture local knowledge of population characteristics and identify local needs; focus group discussions were held with clients to identify existing barriers to implementation, as well as brainstorming sessions with health care providers, PLHIV groups, civil

| Exploration | Preparation | Implementation | Sustainment |
|---|---|---|---|
| Identified existing gaps using program data, AKAIS* and NAIIS report.<br><br>Identified key stakeholders including clients, health care providers, PLHIV groups, civil society groups, community leaders, government officials, donors, implementing partners, etc.<br><br>Key informant interviews with community leaders<br><br>Focused Group Discussion with clients<br><br>3-day brainstorming sessions with relevant stakeholders<br><br>Community scoping visits | Identified potential barriers (e.g. stigma and discrimination, low HIV risk perception, poverty, religious beliefs, risky sexual behaviors, hard-to-reach terrain, poor health-seeking behavior)<br><br>Developed a roadmap for treatment *saturation (Surge Technical Brief)*<br><br>Developed organogram with well-defined roles and responsibilities<br><br>Set up a cluster system for operational efficiency<br><br>Recruited and trained the community ART management (CAM) teams<br><br>Commenced community mapping and development of micro plans<br><br>Set up structures for weekly monitoring (*e.g. ECHO rooms*) and reporting<br><br>Set up a situation room<br><br>Developed a logistics plan<br><br>Developed a strategic behavioral model | HIV Risk stratification<br>Client-centric implementation<br>Context specific strategies<br>Monitoring and feedback<br>Adaptations and adjustments<br>Stakeholders' involvement | • Ongoing support<br>• Continued monitoring<br>• Identification and response to new threats<br>• Scale up interventions that are high-yielding<br>• Continuous stakeholders' involvement & information sharing |

**Fig 1. Adapting EPIS framework for the ART surge implementation in Akwa Ibom, Nigeria (April 2019–March 2021).** *AKAIS: Akwa Ibom AIDS Indicator Survey.

society groups, community leaders, government officials, donors, and implementing partners, to identify Evidence-Based Practices (EBPs) to be adapted for the surge implementation.

The merits and demerits of each EBP were considered based on implementation experience and adapted following established approaches [13]. Table 1 shows the EBPs that were implemented.

**Table 1. Evidence-based practices implemented as SIDHAS Strategies during the surge in Akwa Ibom, Nigeria (April 2019–March 2021).**

| Evidence-Based Practices | SIDHAS Strategies |
|---|---|
| **Evaluative and iterative strategies:** Assess for readiness, identify barriers and facilitators, and conduct a local needs assessment<br>Obtain and use patients'/ consumers' and family feedback Develop and implement tools for quality monitoring, organize quality monitoring systems and purposefully re-examine the implementation. Conduct cyclical small tests of change | Key informant interviews, focus group discussions, brainstorming sessions, community scoping and micro-planning were done<br>Client satisfaction survey (CSS), feedback from clients interrupting treatment, medication adherence checklists, etc. were used to design flexible clinic schedules<br>Data Quality (DQA) and Continuous Quality Improvement (CQI) Assessments were done. Weekly technical sessions with LGA teams and quarterly/ annual strategy review meetings were done. CQI Checklists, 90-day adherence calendar, Interval checklists, etc were developed. Plan Do Study Act (PDSA) cycle was routinely used |
| **Adapt and tailor to context:** Tailor strategies, Promote adaptability | Locally derived solutions such as Creek testing, Camp testing, Batch-up mechanism, Courier services, Community Prevention of Mother to Child Transmission (PMTCT) intervention, Testing at patent medicine stores (Spoke testing), Moonlight and sunrise testing were introduced |
| **Provide interactive assistance:** Provide local technical assistance and clinical supervision | A cluster system was implemented, and service providers from the local communities were recruited. Community teams had Clinicians embedded |
| **Develop stakeholder interrelationships:** Identify and prepare champions, Organize clinician implementation team meetings, Build a coalition, Obtain formal commitments, Visit other sites and develop an implementation glossary | Recruited and trained champions in adolescent care, viral load, index testing, tuberculosis, etc. We conducted daily situation room meetings, CQI meetings, etc. and built several coalitions including the Community Advisory Committee (CACOM), other Implementing Partners, etc. This led to a user fee waiver for PLHIV and COVID-19 movement restriction exemption. Routine peer-to-peer visits were also conducted. Knowledge repository developed |
| **Train and educate stakeholders** | Conducted continuous training, provided continuous consultation, developed, and distributed educational materials, made training dynamic, used train-the-trainer strategies |
| **Engage consumers:** Intervene with patients/consumers to enhance uptake and adherence, Prepare patients/ consumers to be active participants, Use mass media | Linkage to support groups, 90-day adherence calendar, interval checklist, pre-appointment calls and text reminders, peer support using expert clients, Structured age-appropriate counselling, Operation Triple Zero, Literacy programs, use of social and behavioural change materials |
| **Utilize financial strategies:** Place innovation on a fee for service lists/formularies, alter incentive/allowance structures, Alter patient/consumer fees | Community Pharmacy ARV Refill Program, Voucher based ICT, Community DDD models, Home delivery of medication, Elimination of User Fees |
| **Change Infrastructure:** Change record systems, physical structure and equipment, service sites and accreditation or membership requirements | A remote sample login system was introduced, the PCR Laboratory was upgraded, procurement of higher capacity centrifuges, Clients' devolvement to DDD sites, Review criteria for multi-month dispensing, and DDD devolvement |

Engagements with the state government led to the removal of user fees—removing an obstacle for HIV-positive clients—and secured the purchase of additional commodities (including HIV rapid test kits) using local resources to support the surge response. A geographical cluster operational structure was adopted to decentralize technical assistance to the communities. Three clusters, Oron, Eket, and Uyo were formed with six(6), seven(7) and eight(8) LGAs respectively, based on proximity and sociocultural similarities. Each LGA had between 1–7 CAM teams, with each team consisting of 2–3 case finding and tracking teams, creating a team of teams. In each LGA, all CAM teams were linked to one facility, which served as the hub. The clusters were operationally independent in strategy formulation and operational direction [14]. This approach allowed for easy adaptation to the local environment, reduced travel time and time for decision making, created a bottom-top approach to strategy formulation, and ensured close support and coordination to the teams. A central situation room with electronic dashboards was established for data collation, triangulation, and decision-making to provide close monitoring of the surge progress. Daily situation room meetings were used to provide feedback, co-design corrective measures, and provide real-time guidance for early course correction.

CAM teams, linked to health facilities in a hub-and-spoke model for resupplies and reporting, provided comprehensive ART services within the communities to clients who are unable or unwilling to go to health facilities, with clinicians providing clinical supervision [15]. Service providers from the local communities were recruited and trained for this purpose and integrated into these teams to improve community ownership. Facilities were designated into tiers 1, 2, and 3 based on the number of PLHIV on ART (client volume) at each site at the beginning of the surge. We applied the Pareto principle [16] to identify 20% of the sites responsible for 80% of client volume and categorized these as Tier 1 facilities — also referred to as Enhanced Site Management (ESM) facilities [17]. These ESM facilities were provided with additional support (onsite project technical staff providing direct service delivery and mentorship to government workers, frequent supporting visits by supervisory staff, more frequent data reviews, etc.) to improve their performance. The implementation strategies were periodically examined using quality monitoring tools to adjust and refine them for better output.

*Interventions to achieve "95% of all PLHIV know their HIV status".* Table 2 shows the specific interventions that were implemented for HIV case finding. In the health facilities,

**Table 2. HIV case-finding strategies and approaches used during the pre-surge and surge periods.**

| Pre-surge (April 2017–March 2019) | Surge (April 2019–March 2021) |
|---|---|
| | Testing modality |
| Provider Initiated Counselling and Testing | Provider Initiated Counselling and Testing |
| Facility-Based Index Testing | Index Testing (facility and community) |
| | Testing approaches based on-site or time of testing |
| | Community Hotspot testing |
| | Community-Based Index Testing including Voucher-Based Index case testing and use of Genealogy-focused community team (gCAM) |
| | Moonlight Testing |
| | Sunrise Testing |
| | Creek/marine testing |
| | Spoke testing |
| | Self-testing |
| | Camp testing |

Provider Initiated Testing and Counselling (PITC) with risk screening was offered at child health, family planning, malnutrition, and inpatient clinics, and without risk screening at tuberculosis and antenatal clinics. Under Index Case Testing (ICT), sexual partners of newly diagnosed HIV positive individuals or those with unsuppressed VL were invited and offered HIV testing using partner delivered vouchers or contacted by phone after anonymous listing by their partners. Children of HIV-positive women were also enumerated and tested. HIV self-testing was also offered to those who wanted anonymity.

In the community, testing locations were selected based on the local understanding of the social-behavioral characteristics of the communities, discussion with local leaders, joint micro-planning, and geographic information (GIS) guided or geo-targeting of hotspots. Testing services were offered at times that were context-specific, e.g. at night or early morning to reach people like farmers and fisherfolks who were unavailable during the conventional working hours *(moonlight and sunrise testing)* and at different locations such as creeks (creek testing), private laboratories, patent medicine vendors *(spoke testing)*, and traditional birth homes. CAM teams also accessed and camped in distant hard-to-reach communities to provide HTS *(camp testing)* [17, 18].

*Interventions to achieve "95% of those diagnosed with HIV infection were on sustained ART"*. We implemented same-day ART initiation, client navigation services for linkage to ART, structured age-appropriate counseling, support group enrollment, and other peer support services using expert clients. Six Decentralized Drug Distribution (DDD) models–Community Pharmacy ART Refills Program (CPARP), fast track, adolescent refill clubs, community ART refill clubs and groups, and home delivery–were introduced [19, 20], and clients received multimonth dispensing (MMD) with refill frequencies of 3, 3–5 and 6-monthly based on national guidance in any of the models [12, 19, 21]. More adherent clients ultimately received 6 monthly refills. Offering services during weekends and after-hours accommodated those who were unavailable at conventional hours due to work. A 90-day adherence calendar, interval checklist (a series of prompts for case managers to ensure that clients receive appropriate care), pre-appointment calls, text message reminders, and tracking of missed appointments helped providers to monitor clients in care. Access to the DDD models was expanded during the COVID-19 pandemic.

*Interventions to achieve "95% of those on ART are virally suppressed"*. We conducted treatment literacy campaigns and included undetectable = untransmissible ("U = U") messaging using posters and other communication channels within and outside the facilities. To facilitate VL sample collection, a list of eligible clients was generated from the electronic medical records that included phone numbers and residential addresses for follow-up. Sample collection times were staggered based on client preferences and included early morning (sunrise) or late night (moonlight) collection. Where needed, local courier services were used to transport VL samples to the health facilities for processing. Dried Blood Spot testing (DBS) for VL was preferred in hard-to-reach communities.

To ensure all samples were tested promptly, we remotely logged the samples on the online Laboratory Information Management System (LIMS) and transported all samples to the mega Polymerase Chain Reaction (PCR) lab, equipped with seven high throughput PCR machines, working on a 24-hour schedule. Results were printed directly from LIMS at the health facilities. We categorized VL results into unsuppressed (VL $\geq$1000 copies per ml), detectable suppression (40–999 copies per ml), and undetectable (<40 copies per ml) and all clients with VL $\geq$1,000 received enhanced adherence counseling (EAC) in person or virtually through phone calls. SMS reminders were sent to prompt clients to take their medicines; and viral load test was repeated after three (3) months of EAC. At the peak of the COVID-19 pandemic, virtual adherence monitoring was expanded.

### Study design

The study was a retrospective cohort study.

### Data collection

The SIDHAS project leveraged the national and PEPFAR data management platforms–District Health Information System (DHIS); Data for Accountability, Transparency and Impact (DATIM); and Lafiya Management Information System (LAMIS) [22] to report routine program data in aggregate and at patient-level [23] Non-routine program data were collected using Microsoft Excel during the surge.

Data were reported daily by trained data-entry clerks (DEC) into national service registers andLAMIS. These platforms were routinely validated, summarized into national Monthly Summary Forms (MSFs), and transcribed into DHIS and DATIM. Data from these sources were used to monitor project performance during the surge.

Data were abstracted for the pre-surge period (April 2017 –March 2019) and surge period (April 2019 –March 2021) for required variables (HIV testing and positivity, retention, and viral suppression). The abstracted data did not contain any patient identifier.

The key outcomes assessed include HIV testing (Number of individuals provided with HTS), positivity rate (proportion of individuals who received HIV testing that were diagnosed HIV positive), retention (number of individuals alive and on treatment 12 months after ART commencement) and viral load suppression (plasma VL <1,000 copies/ml).

### Data analysis

Descriptive statistics were used to summarize key outcomes by age, gender, and LGA. Interrupted time series analysis (ITS) segmented regression analysis was used to estimate the impact of the surge intervention on HIV testing uptake and numbers tested positive for HIV.

Independent samples Mann-Whitney U test was conducted to compare retention and viral load suppression across the two periods. Retention was determined at 12 months after ART commencement. Clients who missed a clinic appointment and did not return 28 days after expected clinic were considered "not retained" at 12 months [24]. All analyses were performed using SPSS version 26 and a p-value set at 0.05.

### Ethical considerations

This study was reviewed by the Protection of Human Subjects Committee at FHI 360 (Project no 1770609–1) and was determined to be non-human subject research. The authors had no access to the patients or any personally identifying information for the individuals who were included in the study.

## Results

From April 2017 to March 2021, a total of 2,508,532 people were tested for HIV, with 123,444 (4.9%) people diagnosed with HIV (Table 3). There was a significant increase in the number of persons tested from 490,450 to 2,018,082 and in HIV-positive individuals identified from 21,234 to 102,165 during the surge. The median number of people tested for HIV each month increased from 19,444 (17,085–27570) pre-surge to 49,023 (43,312–54,497) during the surge, while the median number of HIV-positive individuals newly diagnosed increased from 906 (755–1197) pre-surge to 2,682 (2,126–2,923) per month during the surge. The yield from HIV testing increased from 4.3% to 5.1% in spite of the dramatic increase in testing volume.

**Table 3. Pre-surge and surge HIV case finding April 2017 to March 2021, Akwa Ibom State, Nigeria.**

| Characteristics | Pre surge (April 2017–March 2019) | | | Surge (April 2019–March 2021) | | |
|---|---|---|---|---|---|---|
| | Counseled, Tested and Received HIV test Results (n) | HIV Positive (n) | Positivity Rate (%) | Counseled, Tested and Received HIV test Results | Tested HIV Positive (n) | Positivity Rate (%) |
| Overall | 490,450 | 21,234 | 4.3% | 2,018,082 | 102,165 | 5.1% |
| Age group | | | | | | |
| 1–4 | 48,976 | 647 | 1.3% | 26,570 | 1012 | 3.8% |
| 5–9 | 22,260 | 362 | 1.6% | 19,883 | 729 | 3.7% |
| 10–14 | 15,518 | 206 | 1.3% | 20,107 | 716 | 3.6% |
| 15–19 | 41,215 | 690 | 1.7% | 137,791 | 3296 | 2.4% |
| 20–24 | 77,647 | 2,586 | 3.3% | 303,915 | 10,052 | 3.3% |
| 25–49 | 247,891 | 15,250 | 6.2% | 1,375,209 | 77,862 | 5.7% |
| 50+ | 36,943 | 1,493 | 4.0% | 134,607 | 8,498 | 6.3% |
| Sex | | | | | | |
| Males | 205,336 | 7,570 | 3.7% | 879,888 | 40,829 | 4.6% |
| Females | 285,114 | 13,664 | 4.8% | 1,138,194 | 61,336 | 5.4% |
| Regions | | | | | | |
| Uyo cluster | 184,339 | 7,591 | 4.1% | 613,537 | 28,285 | 4.6% |
| Oron cluster | 115,193 | 4,254 | 3.7% | 685,065 | 34,265 | 5.0% |
| Eket cluster | 40,065 | 2,571 | 6.4% | 719,480 | 39,615 | 5.5% |
| Testing modalities | | | | | | |
| Facility | 490,450 | 21,234 | 4.3% | 412,419 | 14,565 | 3.5% |
| Community | | | | 1,605,663 | 87,600 | 5.5% |
| Testing strategy | | | | | | |
| Provider Initiated Counselling and Testing | 484,794 | 20,873 | 4.3% | 412,419 | 14,565 | 3.5% |
| Index testing (fac/com) | 2554 | 311 | 12.2% | 132815 | 32069 | 24.1% |
| Antenatal clinic | 3102 | 50 | 1.6% | 89475 | 1286 | 1.4% |
| Creek/marine testing* | | | | 119872 | 11009 | 9.2% |
| Spoke testing* | | | | 7,268 | 430 | 5.9% |
| Self-testing* | | | | 1,759 | 598 | 34.0% |
| Camp testing* | | | | 34579 | 4112 | 11.9% |
| Voucher-Based Index case testing* | | | | 756 | 456 | 60.3% |
| Community Hotspot Testing* | | | | 1,219,139 | 37,640 | 3.1% |

*These models were only introduced during the surge period

Based on the segmented regression analysis, the number of persons tested for HIV decreased significantly by 406 every month prior to surge (95% CI = [-728.7, -82.7], p = 0.015). In the first month of the surge, total number of persons tested for HIV increased significantly by 56,738 (95% CI = [41151.1, 72324.1], p< 0.0001), followed by a significant increase in the monthly trend (relative to the pre-intervention trend) by 1448 in number of persons tested for HIV (95% CI = [138.5, 2757.4], p = 0.031) (Table 4, Fig 2).

With regards to the positive case identification, number of persons tested HIV positive decreased significantly by 22 every month prior to surge (95% CI = [-41.5, -2.4], p = 0.028). In the first month of the surge, total number of persons tested positive increased significantly by 2,600 (95% CI = [1912.9, 3286.9], p< 0.0001), followed by a significant increase in the monthly trend (relative to the pre-intervention trend) by 113 in number of persons tested HIV positive (95% CI = [48.0, 178.3], p = 0.001).

**Table 4. Results from segmented regression analysis on uptake of HIV testing.**

| Variables | Coeficient | 95% C.I | p-value | Coeficient | 95% C.I | p-value |
|---|---|---|---|---|---|---|
| | i. Tested | | | ii. Tested Positive | | |
| Monthly change in number of persons tested, April 2017 -March 2021 | −405.7 | −728.7, −82.8 | 0.015 | −22.0 | −41.5, −2.4 | 0.028 |
| Change in the level of number of persons i. tested and ii. Tested positive | 56,738 | 41151.1, 72324.1 | <0.001 | 2599.9 | 1912.9, 3286.9 | <0.001 |
| Change in trend in monthly number of persons i. tested and ii. Tested positive between April 2019 and March 2021 compared to April 2017.- March 2019 | 1448 | 138.5, 2757.4 | 0.031 | 113.1 | 48.0, 178.3 | 0.001 |

Of the positives identified, 99.2% (n = 21,073) were initiated on ART prior to surge; and 98.2% (n = 100,393) during the surge (Table 5). The median age at ART start was 32 years [26-40years] and 34 years [27-40years] respectively across the two periods. Across the two time periods, the highest number of newly diagnosed individuals were asymptomatic HIV (WHO Clinical Stage I). However, the proportion of asymptomatic individuals was much higher during the surge compared to pre-surge (89.0% vs 43.4%). The proportion of newly diagnosed individuals who started ART on the same day was significantly higher during the surge period compared to the pre-Surge period [98.5% vs 74.1%; p = <0.001].

Fig 3 shows that there was a marked increase in treatment initiation during the surge period, with a sharp decline in ART initiation to below pre-surge levels between March and May 2020, coinciding with the onset of the COVID-19 pandemic in the state.

As shown in Table 6, retention improved overall from 83.3% during the pre-surge period to 92.3% during the surge period (Table 5), increasing more in the 10–14 years age group (12.6% increase; p = 0.001), among males (11.3% increase, p<0.001), Eket cluster (17.0%, <0.001), and in non -ESM sites (10.4% increase, p<0.001).

Table 7 shows viral suppression data before and during the surge. Overall, the viral suppression rate improved significantly from 73.5% to 96.2%. This improvement was highest among children 10–14 years (52.3% increase, p<0.001), and lowest in older adults 45–49 years (17.5% increase).

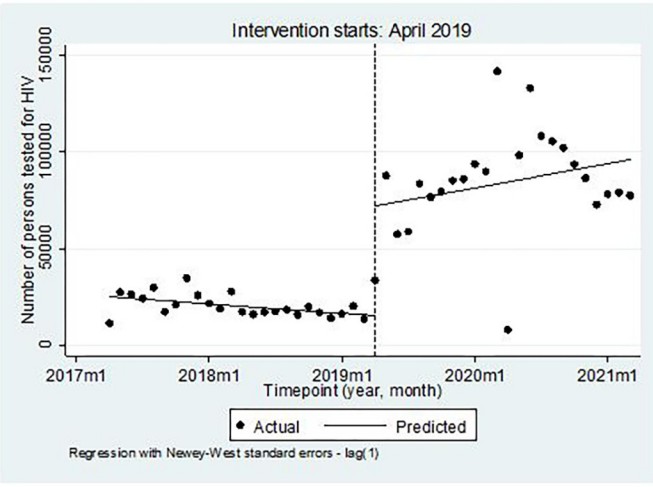
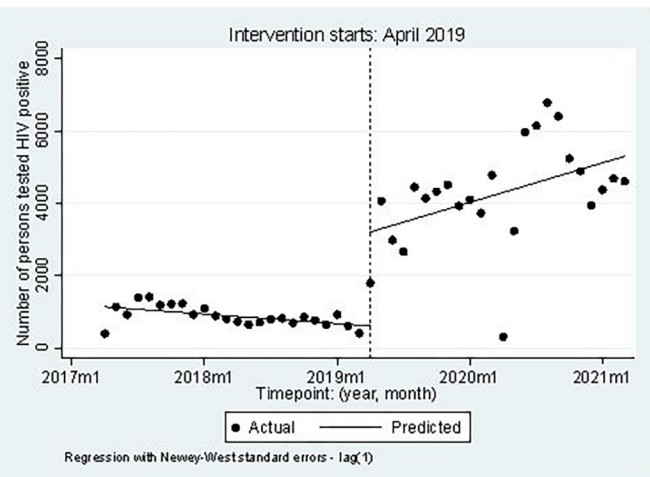

**Fig 2. Actual and predicted trends in total number of persons i. tested and ii tested positive, by month, April 2017–March 2021.**

**Table 5. Demographic and clinical characteristics of individuals initiated on ART before and during the surge, Akwa Ibom, Nigeria (April 2017 to March 2021).**

|  | Pre-surge (April 2017–March 2019) | Surge (April 2019–March 2021) | P-value |
|---|---|---|---|
| PLHIV diagnosed and started ART | 21,073 (99.2%) | 100,393 (98.3%) | <0.001 |
| Sex |  |  |  |
| Females | 14,864 (91.9%) | 60,860 (99.2%) | <0.001 |
| Males | 6,209 (82.0%) | 39,533 (96.8%) | <0.001 |
| Median (interquartile range) Age (years) n = 121,466 | 32 (26–40 years) | 34 (27–40 years) |  |
| WHO staging n = 120,384 |  |  |  |
| Stage 1 | 8,887 (43.4%) | 88,836 (89.0%) |  |
| Stage 2 | 5,735 (28.0%) | 9,327 (9.3%) | <0.001 |
| Stage 3 | 5,534 (27.0%) | 1,616 (1.6%) |  |
| Stage 4 | 342 (1.7%) | 107 (0.1%) |  |
| ART initiation |  |  |  |
| Same day | 16,185 (74.1%) | 96901 (98.5%) | <0.001 |
| 1–14 days | 2,884 (13.2%) | 1518 (1.5%) |  |
| >14 days | 2,772 (12.7%) | 0 (0%) |  |

## Discussion

The implementation of a combination of specific programmatic and technical interventions during the Akwa Ibom surge was associated with a marked increase in HIV case finding and

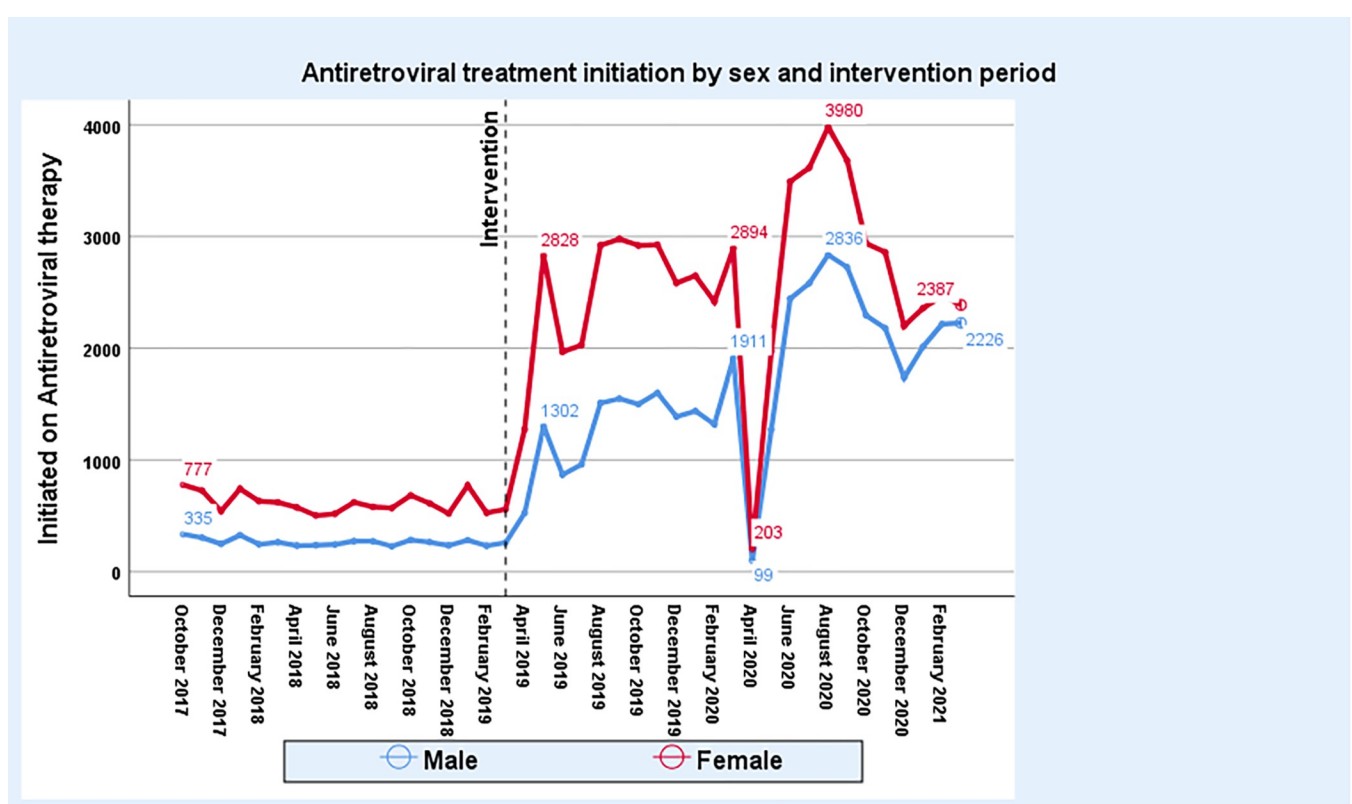

**Fig 3. Number of people initiated on ART October 2017-February 2021.**

**Table 6. Retention by intervention period and key client characteristics (April 2019–March 2021).**

| Characteristics | Pre-surge commencement | | Surge commencement | | | |
|---|---|---|---|---|---|---|
| | *n* | *retention rate* | *n* | *retention rate* | *Percentage change* | *p-value* |
| Overall | 21,073 | 83.3% | 100,393 | 92.3% | 15% | <0.001 |
| Age groups | | | | | | |
| 0–9 | 901 | 81.9% | 1772 | 81.0% | -1.1% | 0.585 |
| 10–14 | 16 | 78.5% | 620 | 89.8% | 12.6% | <0.001 |
| 15–19 | 688 | 77.8% | 3364 | 86.3% | 9.8% | <0.001 |
| 20–24 | 2,529 | 79.0% | 10,099 | 89.0% | 11.2% | <0.001 |
| 25–29 | 4,061 | 82.3% | 17,793 | 92.1% | 10.6% | <0.001 |
| 30–34 | 3,983 | 84.2% | 19,336 | 92.8% | 9.3% | <0.001 |
| 35–39 | 1,557 | 85.0% | 9,459 | 95.0% | 10.5% | <0.001 |
| 40–44 | 2,137 | 85.7% | 12,397 | 93.9% | 8.7% | <0.001 |
| 45–49 | 1,506 | 85.9% | 8,671 | 93.8% | 8.4% | <0.001 |
| 50+ | 3,548 | 84.6% | 16,882 | 92.5% | 8.5% | <0.01 |
| Gender | | | | | | |
| Females | 14,864 | 83% | 60,860 | 91.4% | 9.1% | <0.001 |
| Males | 6,209 | 83% | 39,533 | 93.6% | 11.3% | <0.001 |
| Care setting | | | | | | |
| Facility | 21,061 | 83.3% | 14,342 | 89.6% | 7.0% | <0.001 |
| Community | - | - | 86,051 | 92.7% | | - |
| Cluster | | | | | | |
| Uyo cluster | 11,447 | 85% | 28,754 | 86.3% | 1.5% | 0.114 |
| Oron cluster | 7,958 | 81% | 33,532 | 95.3% | 15.0% | <0.001 |
| Eket cluster | 1,668 | 78% | 38,107 | 94.0% | 17.0% | <0.001 |
| Site category | | | | | | |
| ESM (>1,500 clients per site) | 17,361 | 83.9% | 78,978 | 92.9% | 9.7% | <0.001 |
| Non-ESM | 3,712 | 80.3% | 21,415 | 89.6% | 10.4% | <0.001 |

significant improvements in retention and viral suppression. Within 24 months of the surge implementation, a total of 2,018,082 persons were tested for HIV (representing 37% of the state population), and 102,165 people living with HIV were diagnosed and initiated on treatment. The advent of the COVID -19 pandemic in April 2020 had little effect on the surge trajectory as the program introduced several adaptations to overcome the movement restrictions, physical distancing, and other infection control measures. Of note, the majority of those diagnosed during the surge were largely asymptomatic.

PEPFAR programs have used various strategies across different countries to close antiretroviral therapy gaps. For example, a US Centers for Disease Control and Prevention (CDC)—led surge implemented across nine states in Nigeria for 18 months (May 2019–September 2020) led to an eightfold increase in the number of people newly diagnosed with HIV and a 65% increase in the total number of persons receiving ART [25]. In Mozambique, surge implementation was associated with a 40% increase in case finding while in Malawi case finding increased by 51.9% [18, 26]. We achieved comparable results by applying a combination of high volume-low yield, and low volume-high yield testing strategies mostly in the community, and introduced an innovative Community ART Management (CAM) approach for timely testing, linkage, and support to clients on ART. Our findings of a significant increase in HIV case identification, of mostly asymptomatic individuals provide more evidence that using a mix of approaches can address the unmet need for ART and help close treatment gaps, and have potential prevention implications.

**Table 7. Participant characteristics and viral suppression by intervention periods.**

| Characteristics | Pre-surge commencement | | Surge commencement | | | |
|---|---|---|---|---|---|---|
| | *number with viral load result* | *suppression rate* | *number with viral load result* | *suppression rate* | Percentage change | *p-value* |
| Overall | 13,958 | 73.5% | 73,587 | 96.2% | 23.6% | <0.001 |
| Age groups | | | | | | |
| 0–9 | 525 | 45.1% | 1,188 | 90.2% | 50.0% | <0.001 |
| 10–14 | 107 | 43.95% | 479 | 92.1% | 52.3% | <0.001 |
| 15–19 | 352 | 66.8% | 2,650 | 93.8% | 28.8% | <0.001 |
| 20–24 | 1,498 | 71.0% | 7,786 | 95.4% | 25.6% | <0.001 |
| 25–29 | 2,677 | 73.9% | 13,286 | 96.5% | 23.4% | <0.001 |
| 30–34 | 2,711 | 73.9% | 13,954 | 96.4% | 23.3% | <0.001 |
| 35–39 | 1,073 | 76.2% | 12,792 | 97.0% | 21.4% | <0.001 |
| 40–44 | 1,526 | 75.2% | 8,899 | 96.8% | 22.3% | <0.001 |
| 45–49 | 1,058 | 79.9% | 6,258 | 96.9% | 17.5% | <0.001 |
| 50+ | 2,431 | 76.9% | 6,295 | 96.2% | 20.1% | <0.001 |
| Gender | | | | | | |
| Male | 4209 | 72.1% | 28,184 | 96.6% | 25.4% | <0.001 |
| Female | 9,749 | 74.2% | 45,403 | 95.9% | 22.6% | <0.001 |
| Care setting | | | | | | |
| Facility | 13,957 | 66.3% | 10,560 | 96.7% | 31.4% | <0.001 |
| Community | - | - | 63,027 | 96.1% | | - |
| Geographic area | | | | | | |
| Uyo cluster | 8,704 | 77.4% | 19,877 | 92.7% | 16.5% | <0.001 |
| Oron cluster | 41,097 | 66.0% | 25,793 | 96.2% | 31.4% | <0.001 |
| Eket cluster/ | 1,145 | 70.9% | 27,917 | 98.5% | 28.0% | <0.001 |
| Facility Setting | | | | | | |
| ESM | 12,077 | 74.3% | 58,119 | 96.3% | 22.8% | <0.001 |
| Non-ESM | 1,881 | 68.4% | 15,468 | 95.6% | 28.5% | <0.001 |

Our results demonstrate the efficiency of the HIV testing strategies during the surge. Index case testing–conventional and voucher-based, with positivity yields of 24% and 60% respectively, emerged as the most efficient approach in HIV case finding during the surge period. This aligns with studies conducted in Tanzania, Lesotho, and South-West Nigeria, suggesting the potential advantage of this testing approach in the future as new HIV cases become more difficult to identify [27–29]. Index testing prioritizes identifying sexual partners of those who are newly diagnosed, virally unsuppressed, and biological children at imminent risk of HIV [12].

The facility-based testing yield reduced from 4.3% pre-surge to 3.5% during the surge. Deployment of community testing helped identify more individuals with previously undiagnosed HIV who were largely asymptomatic. Effective targeting of geographies with high-risk individuals and timing of testing helped achieve a positivity rate of 5.5% at the community level. We maintained a high positivity yield throughout the surge period despite the increase in testing volume. This was achieved through a mixture of strategies, data review, and the introduction of HIV risk screening before HIV testing which ensured that only high–risk individuals in the community were tested. Other studies have reported high HIV testing efficiency through risk screening before testing [30–32]. At a time of declining resources, efficient testing approaches are encouraged and in line with the WHO recommendations of targeted HIV testing using a symptom screening approach in the general population [33].

Some studies have highlighted that early HIV identification and treatment initiation are associated with a decline in disease progression, reduction in HIV transmission, and HIV-related mortality [34, 35]. Our results show that 89% of PLHIV diagnosed during the surge were in WHO clinical stage I, compared to 43% pre-surge. We believe this was due to prioritizing index testing and targeted community testing that were informed by extensive planning and engagement of staff with thorough knowledge of the localities. Similarly, the number of people who were offered and accepted same-day ART initiation increased from 74.1% pre-surge to 98.5% during the surge. This could be partly explained by the National HIV/AIDS treatment guideline [12] change that recommended same-day ART in December of 2016 but could also be due to providing ART in the community close to where people reside.

In the cohort studied, retention increased from 83% before the surge to 98% during the surge period. While no specific intervention can explain this increase, our approaches ensured that an increasing cohort of patients was accommodated within the program using differentiated service delivery (DSD) models that were implemented to adapt services to client needs [19], as well as a high-frequency monitoring system that identified and flagged missed appointments for early follow-up, including recovery and re-engagement with clients who had dropped off from the treatment program. We observed poor retention among children in the 0–9 years age group that requires further exploration. Most of the DSD models described by Sanwo et al. [19] were predominantly for adults and did not specifically target the unique service delivery needs of children living with HIV (CLHIV). Future models of care should consider the complexities of providing care to CLHIV [36], and thus, be designed to optimize outcomes for this important sub-population.

The Akwa Ibom surge strategies led to a 19% increase in viral suppression, which is higher than the 1% achieved in the Mozambique surge [26]. Virtual approaches such as tele-EAC (using phone calls and SMS reminders) were introduced at the peak of the COVID-19 pandemic to ensure continuity in client care and adherence monitoring. Unlike retention, the highest improvement in VL suppression was observed in children. This improvement may be due to a quality improvement initiative that addressed identified root causes such as adherence counseling and regimen optimization. The introduction of TLD in 2018 and scale up of other DTG-based regimen during the surge could have contributed significantly to this increased viral suppression.

Our study had some limitations. Our analysis covered 21 out of 31 LGAs in the state so the results and conclusions derived may not apply to the entire state. Secondly, several strategies were deployed concurrently during the surge but this study did not attempt to determine the individual contribution of strategies to overall program outcome.

Despite the limitations, our study had major strengths. First, we compared two different implementation periods of 24 months each, generating several months of program follow up which allowed us to evaluate retention and viral suppression outcomes. Most studies that have assessed surge implementation have had short follow-up periods and lacked a comparison period. Most followed up implementation for between 6 weeks to 18 months [18, 25, 26]. Secondly, we reviewed program outcomes across the HIV continuum of care, thus, allowing a more comprehensive evaluation of interventions during the HIV surge in the state.

## Conclusions

We described the components of the Akwa Ibom surge intervention, and determined the collective contribution of these interventions toward improving HIV case-finding, and other program outcomes across the HIV continuum of care. The surge implementation led to the development and adoption of evidence-based practices that increased the number of PLHIV

on ART, those retained in the program, and virally suppressed. These results have implications for program implementers who could adapt a surge approach and a blend of technical interventions delivered backed by consultative programming to achieve program aims or show feasibility. Further studies will be required to examine the costs of intervention.

## Supporting information

**S1 Data. Pre surge and surge data (disaggregated).** Brief description of file content: Sugre Counselled Tested and Receive Result (CTRR) vs Positive. Information on the file format: Microsoft Excel Workbook.
(ZIP)

## Acknowledgments

The authors acknowledge all those who were involved in the SIDHAS project in Nigeria, particularly the technical and strategic information staff members based at the various facilities and the clinicians leading the community ART management teams.

## Author Contributions

**Conceptualization:** Pius Nwaokoro, Helen Idiong, Philip Imohi, Dorothy Oqua, Kunle Kakanfo, Kolawole Olatunbosun, Augustine Umoh, Moses Bateganya.

**Data curation:** Uduak Akpan, Esther Nwanja, Bala Gana, Titilope Badru.

**Formal analysis:** Uduak Akpan, Bala Gana, Titilope Badru, Augustine Idemudia.

**Methodology:** Pius Nwaokoro, Otoyo Toyo, Esther Nwanja, Titilope Badru, Moses Bateganya.

**Resources:** Pius Nwaokoro, Kufre-Abasi Ukpong, Echezona Ezeanolue.

**Supervision:** Olusola Sanwo, Helen Idiong, Matthew-David Ogbechie, Philip Imohi, Anthony Achanya, Dorothy Oqua, Kunle Kakanfo, Kolawole Olatunbosun, Augustine Umoh, Patrick Essiet, Ime Usanga, Chika Obiora-Okafo, Ezekiel James, Isa Iyortim, Robert Chiegil, Hadiza Khamofu, Satish Raj Pandey, Moses Bateganya.

**Validation:** Otoyo Toyo, Titilope Badru, Moses Bateganya.

**Visualization:** Uduak Akpan, Bala Gana, Titilope Badru.

**Writing – original draft:** Pius Nwaokoro, Otoyo Toyo, Uduak Akpan, Esther Nwanja, Iheanyichukwu Elechi, Kufre-Abasi Ukpong, Augustine Idemudia.

**Writing – review & editing:** Olusola Sanwo, Matthew-David Ogbechie, Philip Imohi, Anthony Achanya, Dorothy Oqua, Kunle Kakanfo, Kolawole Olatunbosun, Augustine Umoh, Patrick Essiet, Ime Usanga, Echezona Ezeanolue, Chika Obiora-Okafo, Ezekiel James, Isa Iyortim, Robert Chiegil, Hadiza Khamofu, Satish Raj Pandey, Moses Bateganya.

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
