## [Decision Letter · Decision Letter 0]

19 Sep 2022

PONE-D-22-14919Achieving HIV Epidemic Control through integrated community and facility-based strategies: Lessons Learnt from ART-surge Implementation in Akwa Ibom, NigeriaPLOS ONE

Dear Dr. Nwaokoro,

Thank you for submitting your manuscript to PLOS ONE. After careful consideration, we feel that it has merit but does not fully meet PLOS ONE’s publication criteria as it currently stands. Therefore, we invite you to submit a revised version of the manuscript that addresses the points raised during the review process.

Two external reviewers have evaluated your submission and have identified several aspects of the study design and methods that require clarification, and have pointed to opportunities to improve the presentation of the manuscript. Please refer to their detailed comments below and respond carefully to all of the points they have raised when preparing your revision.

We look forward to receiving your revised manuscript.

Kind regards,

Jamie Males

Editorial Office

PLOS ONE

Journal Requirements:

This publication resulted in part from data collected during the implementation of the PEPFAR-funded SIDHAS project in Nigeria (Cooperative Agreement Number: AID-620-A-11-00002).

Reviewers' comments:

Reviewer's Responses to Questions

**Comments to the Author**

1. Is the manuscript technically sound, and do the data support the conclusions?

Reviewer #1: Yes

Reviewer #2: Yes

2. Has the statistical analysis been performed appropriately and rigorously? 

Reviewer #1: Yes

Reviewer #2: Yes

3. Have the authors made all data underlying the findings in their manuscript fully available?

Reviewer #1: Yes

Reviewer #2: No

4. Is the manuscript presented in an intelligible fashion and written in standard English?

Reviewer #1: Yes

Reviewer #2: Yes

5. Review Comments to the Author

Reviewer #1: 1. Lines 34 - 36: It was stated that ANC sentinel survey and NAHRS over estimated prevalence and number of PLHIV. The statemen is true for prevalence but not completely accurate for number of PLHIV because these 2 surveys do not directly estimate PLHIV population. PLHIV population is estimated using a model. Please correct the statement.

2. Line 40: The same comment above applies to NAIIS survey.

3. The statement “AkwaIbom ranks highest in HIV prevalence in Nigeria” is a very strong statement and is one of the major findings of NAIIS. I wonder why the authors decided to use online newspaper as reference instead of the primary reference (NAIIS) which is more scientific. Consider using NAIIS report please.

4. Line 67: It will be good to describe the peculiarities of the two phases of the Surge Apr 2019 – Sep 2019 vs Oct 2019 – Mar 2021. Why is it in two phases, why the scale down in No. of LGAs in the 2nd phase? Additionally, the 102 facilities involved are they from both phases or from the 2nd phase only?

5. Line 79: use of risk stratification tool; please qualify as use of HIV risk stratification tool. Additionally, it will be good to define the tool and its purpose or reference where the reader can get details about the tool.

6. Line 83: results were promptly returned to providers through an online platform. This is very important point and I recommend the authors to explain the online platform and its functionality. The objective of this paper is to provide lessons learnt. Therefore, the authors should please not assume that the readers are familiar with everything stated.

7. Line 87: Surge framework – this is an important component in the surge and should have been introduced early. For example, this could have been at the beginning of the subsection – Surge Intervention. Something like “the Surge intervention was based on EPIC framework ……………., then sentence in line 78 can follow. Additionally, the Surge Framework should be introduced in the abstract.

8. Line 92: the sentence “focus group discussions with clients to identify existing barriers to implementation in their...” is hanging, incomplete.

9. Figure 2 caption should be more elaborate.

10. Lines 101 – 105: It will be good to understand how many LGAs per cluster, how many CAM and sub teams. This level of granularity is important for the reader to understand the complexity of the operations deployed for the Surge. Use of “several” may not be adequate

11. Line 97: The title of Table 1 is a bit misleading when compared to the statement where table 1 was referenced. The statement says “The merits and demerits of each EBP were considered based on implementation experience and adapted following established approaches [12]. Table 1 shows the EBPs that were implemented”. Is the EPIC synonymous with the EBP? Consider revising the title and if possible, remove EPIC or introduce EPIC in line 97 to ensure consistency with the table title.

12. Table 1 is too long and could be more concise. Some EBPs could be merged into a single EBPs. Similarly, SIDHAS strategies could also be regrouped based on strategies that are similar thereby shortening the table. Due to length of the table and that the strategies are not systematically presented, this push the reader to be flipping up and down trying to align strategies that are similar in order to contextualize the approach.

13. Line 124: “those with the highest client volume received volume-intensive support”. Phrase not clear.

14. Line 131 and table 2: Reaching 1st 95% - very important points have been raised. It will be appropriate if the authors will provide additional context on how several strategies here were achieved or they should provide references so that readers can get in-depth understanding on how these strategies can be replicated. For example, HIV self-testing, geo-targeting of hotspots, engaging private labs, patent medical vendors TBAs e.t.c.

15. Line 155: “clients received multi month dispensing (MMD) with refill frequencies of 3, 3-5 and 6-monthly in any of the models”. Will be good to describe under what circumstances the the decision to use 3-, 3-5- and 6-month MMDs. Alternatively provide reference.

16. Line 182: Data collection: Since DHIS, DATIM are National and PEPFAR data management platforms, the authors may wish to consider making this clearer by stating that SIDHAS leveraged National and PEPFAR data management platforms, DHIS and DATIM respectively to …….

17. Please correct definition of DATIM from “Data for Accountability, Integrity, and Management” to “Data for Accountability, Transparency and Impact”. Also provide reference.

18. Line 182. Data collection. It will be good for the authors to be very explicit in this section by listing the variables and indicators that were collected. Additionally, DHIS and DATIM were mentioned in the 1st paragraph, but DATIM was never mentioned again. What data goes in to DATIM and what role did DATIM played in this data collection. Additionally, LAMIS was mentioned but then EMR was mentioned later. Please be consistent on whether to use LAMIS or EMR as not all readers may understand both mean the same.

19. Line 197: Please define “key outcomes” by listing them

20. Line 203: This statement is a bit confusing “Clients were considered to be retained in care if their next ART pickup date was after March 31, 2019, for the pre-surge cohort and March 31, 2021, for the surge cohort”. Please provide definition of retention in the context of this study in number of months and also provide starting period for both pre-surge and surge cohort periods.

21. Line 210: Data for this study were collected from an existing project database used for routine program monitoring. Is this database different from DHIS and DATIM earlier stated in data analysis section? If yes, then it may be okay to remove DATIM and DHIS from the data analysis section and to use project database.

22. Results, Table 3: In line 102, Oron, Eket, and Uyo were listed as the only 3 clusters for the Surge. Here you added Ikot Ekpene cluster as additional cluster in pre-Surge. To enable good comparison (pre-Surge and Surge periods) and to minimize confusions, please consider dropping the Ikot Ekpene from Table 3.

23. Results: Table 3 contradicts Table 2 (testing modality and testing strategy). In Table 2, for example community testing was not part of pre-Surge activities but in Table 3 results were provided under community testing. Also, in Table 2, Genealogy and Community Hotspot testing were indicated but no results indicated in Table 3. Please reconcile. If no data for these strategies, then include this under limitations.

24. Results, Table 3: Positivity rates are inconsistently presented to either whole number or 2 decimal percentages. Please present all positivity rates to 1 decimal percentage. This is how prevalence and testing yields are standardly presented.

25. Discussions: Line 298 “This endeavor helped increase the treatment coverage across 21 high burden local government areas and the overall treatment coverage to 87.8%”. These findings were not presented in results section and may not be discussed under the discussions section. If the authors decided to discuss this, then they should indicate LGA level coverage in the results. Additionally, baseline coverage should be presented so that increase in coverage over the 24 months (Surge period) can be appreciated.

26. Discussions: Line 309 “…….and introduced an innovative Community ART Management (CAM) approach for timely testing, linkage, and to provide support to clients on ART”. The sentence seems grammatically in correct, after “for” “to provide” was used. Consider “…….and introduced an innovative Community ART Management (CAM) approach for timely testing, linkage, and support to clients on ART”.

27. Discussions: Line 322 “We improved the impact of this strategy by targeting people who were on treatment but virally unsuppressed”. To aid clarity add “…. the partners of people…” because it is the partners of unsuppressed individuals on ART that are targeted.

28. Additionally, does the result indicate how adding partners of unsuppressed individuals on ART affect the index testing strategy? What is the contribution of partners of unsuppressed individuals to the total positives identified through index testing?

29. Discussions: Line 326 “Effective targeting or geographies with high-risk individuals and timing or testing times helped achieve a high positivity rate of 5.5% at the community level”. Your result (Table 3) shows 5.3% and not 5.5%. Additionally, pre-Surge community testing yield was just 1.0%. I wonder why the authors didn’t mention this comparison here. Although, in my earlier comment (No.23) above, I requested the authors to reconcile Tables 2 and 3 whereby strategies not mentioned in the methods were shown in results and community testing during pre-Surge was one of them.

30. Discussions: Line 328 “We maintained a high positivity yield throughout the surge period through a mixture of strategies, reviewing data, and use of HIV risk screening before HIV testing which ensured that only high–risk persons in the community setting were tested”. The authors may wish to reference a similar study in Nigeria Surge which used the same strategies. This will diversify reader’s understanding of this paper and the strategies described. See: doi: 10.2147/HIV.S316480. eCollection 2021.

Reviewer #2: Nwaokoro and colleagues summarize the impressive programmatic results of a "surge" initiative in Nigeria. Compared to pre-surge, HIV clinical cascade, there were significant and substantial increases in case identification, linkage to ART, same-day ART, retention and viral load suppression -- all in the face of COVID-19. It was also notable that the large increase in cases identified was not simply the result of indiscriminate massive-volume HIV testing, but rather strategic testing mix which resulted in an increase in test positivity even as testing volumes increased. The paper is well organized and clearly written overall.

Major comments:

(1) It would be helpful if the authors included information about the funding levels during the pre-surge and surge periods. How the did the relative increase in funding during surge compare to the relative increases in cases identified and patients with VL suppression?

(2) Did the program scale up TLD and other DTG-based regimens during the surge? This would be a potentially major contributor to the increase in VL suppression during the surge that may should be accounted for in the results and discussion.

(3) There was a large number of interventions and activities implemented as part of the surge. It would be helpful to know if the authors were able to determine which interventions/activities appeared to account for relatively greater impact on clinical outcomes. This may be something that should be added to the limitations.

Minor comments:

(1) p 11. The authors indicate that EAC was provided if the VL was >1000. In that situation, how soon was the VL repeated?

(2) Table 3. The % POS during surge should be 5%, not 4%. In the text of the results (p 13), it would be good to mention that % POS increased even with dramatic increases in testing volumes, an important achievement.

(3) It is even more remarkable that these results were achieved while COVID-19 was raging. Consider adding that context more clearly to the discussion.

6. PLOS authors have the option to publish the peer review history of their article (what does this mean?). If published, this will include your full peer review and any attached files.

Reviewer #1: No

Reviewer #2: No

---

## [Author Response · Author response to Decision Letter 0]

13 Nov 2022

I have responded to each point raised by the academic editor and reviewer(s). - 'Response to Reviewers'.

---

## [Decision Letter · Decision Letter 1]

25 Nov 2022

Achieving HIV Epidemic Control through integrated community and facility-based strategies: Lessons Learnt from ART-surge Implementation in Akwa Ibom, Nigeria

PONE-D-22-14919R1

Dear Dr. Nwaokoro,

We’re pleased to inform you that your manuscript has been judged scientifically suitable for publication and will be formally accepted for publication once it meets all outstanding technical requirements.

Kind regards,

Clement Ameh Yaro, Ph.D

Academic Editor

PLOS ONE

Additional Editor Comments (optional):

Reviewers' comments:

Reviewer's Responses to Questions

**Comments to the Author**

1. If the authors have adequately addressed your comments raised in a previous round of review and you feel that this manuscript is now acceptable for publication, you may indicate that here to bypass the “Comments to the Author” section, enter your conflict of interest statement in the “Confidential to Editor” section, and submit your "Accept" recommendation.

Reviewer #1: All comments have been addressed

Reviewer #2: All comments have been addressed

2. Is the manuscript technically sound, and do the data support the conclusions?

Reviewer #1: Yes

Reviewer #2: Yes

3. Has the statistical analysis been performed appropriately and rigorously? 

Reviewer #1: Yes

Reviewer #2: N/A

4. Have the authors made all data underlying the findings in their manuscript fully available?

Reviewer #1: Yes

Reviewer #2: Yes

5. Is the manuscript presented in an intelligible fashion and written in standard English?

Reviewer #1: Yes

Reviewer #2: Yes

6. Review Comments to the Author

Reviewer #1: The authors have responded to all comments and the manuscript can now be accepted for publications..

Reviewer #2: (No Response)

7. PLOS authors have the option to publish the peer review history of their article (what does this mean?). If published, this will include your full peer review and any attached files.

Reviewer #1: No

Reviewer #2: No

---

## [Editor Report · Acceptance letter]

12 Dec 2022

PONE-D-22-14919R1 

Achieving HIV Epidemic Control through integrated community and facility-based strategies: Lessons Learnt from ART-surge Implementation in Akwa Ibom, Nigeria. 

Dear Dr. Nwaokoro:

I'm pleased to inform you that your manuscript has been deemed suitable for publication in PLOS ONE. Congratulations! Your manuscript is now with our production department. 

Kind regards, 

on behalf of

Dr. Clement Ameh Yaro 

Academic Editor

PLOS ONE